# Integrating Life Cycle Assessment in Conceptual Aircraft Design: A Comparative Tool Analysis

Kristina Mazur *, Mischa Saleh and Mirko Hornung

Chair of Aircraft Design, Technical University of Munich, 85748 Garching, Germany; mirko.hornung@tum.de (M.H.)
* Correspondence: kristina.mazur@tum.de

**Abstract:** Early and rapid environmental assessment of newly developed aircraft concepts is eminent in today's climate debate. This can shorten the decision-making process and thus accelerate the entry into service of climate-friendly technologies. A holistic approach within the conceptual aircraft design is taken into consideration in terms of a life cycle assessment (LCA) to properly model and evaluate these concepts. To provide an understanding of how different LCA software affects the assessment, the goals of this study are to establish a baseline metrics definition for comparative evaluation and apply them to two tools. The first tool is an existing simplified derivative of *openLCA*, while the second, developed in this study, is an automated interface to the same software. The main finding is that researchers and practitioners must carefully consider the intended use of the tool. The simplified tool is suitable for training and teaching purposes and assessments on single score level. In contrast, an advanced tool is required in order to appropriately analyze the overall impact categories requiring high levels of LCA expertise, modeling, and time effort.

**Keywords:** life cycle assessment; conceptual aircraft design; tool analysis; *openLCA*; application programming interface

## 1. Introduction

As the climate debate becomes more prominent in governments and the general public worldwide, research institutions and industries are focusing increasingly on novel ways of assessing the environmental impact of their research topics or products. The aviation industry is no exception to this. While the sector is characterized by steady demand and continually increasing air traffic on the one hand, on the other, it has the highest average greenhouse gas emission contribution per passenger kilometer among the motorized modes of passenger transport (EU-27, 2014–2018 [1]). Additionally, aviation is known for its lengthy development cycle and production process for new configurations. Furthermore, new measures like using hydrogen as a propellant, switching to sustainable aviation fuels, or enhancing airport infrastructure are now being discussed, and require an early environmental impact analysis [2]. This leads to the conclusion that an assessment in the early stage of aircraft design is needed to accomplish the climate targets in the near future. To appropriately model and assess these measures, a holistic approach in terms of a life cycle assessment (LCA) within the conceptual aircraft design is considered.

The methodological foundation of performing a LCA was laid out in the 1990s within the standards ISO 14040 [3] and ISO 14044 [4]. Within these norms, e.g., specific terms are defined, and the general procedure for performing a LCA is explained. Several aircraft assessments have been conducted in accordance with it; however, they exhibit substantial discrepancies in the implementation with respect to the used data, software, and methodology and are therefore comparable only to a limited extent [5–8]. These discrepancies are also visible across industries. To cope with that, initiatives such as the *Life Cycle Initiative* hosted by the UN Environmental Programme are currently extending the guidelines globally. The *Global Guidance on Environmental Life Cycle Impact Assessment Indicators (GLAM)* [9]

and the *Global LCA Data Access (GLAD)* [10] projects aim to, e.g., enhance global consensus on environmental life cycle impact assessment indicators and improve data accessibility and interoperability. These projects aim to eliminate the discrepancies related to the assessment methodology and data. However, there are currently no projects focusing on establishing a guideline for the implementation of and data handling in tools and software itself [11] (According to the FAQ of GLAD, it is a long-term goal and part of the vision to easily import GLAD data to LCA software. However, this is not the case at the moment. Eventually, GLAD should be interoperable with LCA software, perhaps even with access to the database integrated into LCA software tools).

To understand what impact the use of different software can have, this work focuses on the comparative analysis of two LCA tools within the conceptual aircraft design. For that, a methodology is defined, including relevant qualitative and quantitative metrics, e.g., the software robustness or the environmental indicators. Hence, the first research question is:

**RQ1:** What qualitative and quantitative metrics are relevant when comparing different LCA software within the conceptual aircraft design?

The first LCA application used in the analysis was developed by Johanning [12]. The tool, named *LCA-AD* or simplified tool, is an open-source tool derived from the LCA software *openLCA*. It focuses explicitly on comparing potential future concepts, such as hydrogen or electric-powered aircraft. Kossarev et al. [13] have updated the tool with a more current impact assessment method, improved it with a new climate model, and bug-fixed it. The second LCA application, named *openLCA-AD* or advanced tool, was developed within this study and establishes a direct linkage to the previously utilized *openLCA* software. With that, it is aimed to answer the second research question:

**RQ2:** To what extent do different tools affect the assessment of the environmental footprint of future aircraft designs?

To address the research questions, this paper is structured as follows: Section 2 covers the fundamental aspects of life cycle assessment. It begins with a description of the general working principle of a LCA, and proceeds with an overview of current LCA software. Then, LCA is explained with respect to the use case of the conceptual aircraft design. It is followed by a review of previous aircraft LCA studies with a focus on their used software, including a detailed explanation of the simplified *LCA-AD* tool by Johanning [12]. Section 3 then explores the applied software evaluation methodology of this work. It defines the general assumptions and limitations and refers to the **RQ1** establishing metrics the tools are compared with. In Section 4, the implementation of the second tool *openLCA-AD* as an advanced automated LCA interface is presented. With this knowledge, Section 5 provides the LCA results and its discussion in relation to the defined metrics. Thus, the effect of different LCA tools is evaluated to answer **RQ2**. Lastly, Section 6 concludes the findings. The main objectives in this are summarized as:

- A baseline metric definition for comparative evaluation of LCA tools;
- Development and implementation of the automated LCA interface *openLCA-AD*;
- Evaluation of the simplified *LCA-AD* versus advanced *openLCA-AD* tools.

## 2. Fundamentals

### 2.1. LCA in General

Environmental life cycle assessments are used to examine how activities and products affect the environment throughout their entire lives. ISO 14040 [3] is the foundational standard, outlining the principles and framework for conducting LCA. ISO 14044 [4] provides detailed requirements and guidelines for performing LCA studies, including practical recommendations for data collection, impact assessment, and reporting. There are four phases: (A) goal and scope definition, (B) inventory analysis, (C) impact assessment, and (D) interpretation, as shown in Figure 1.

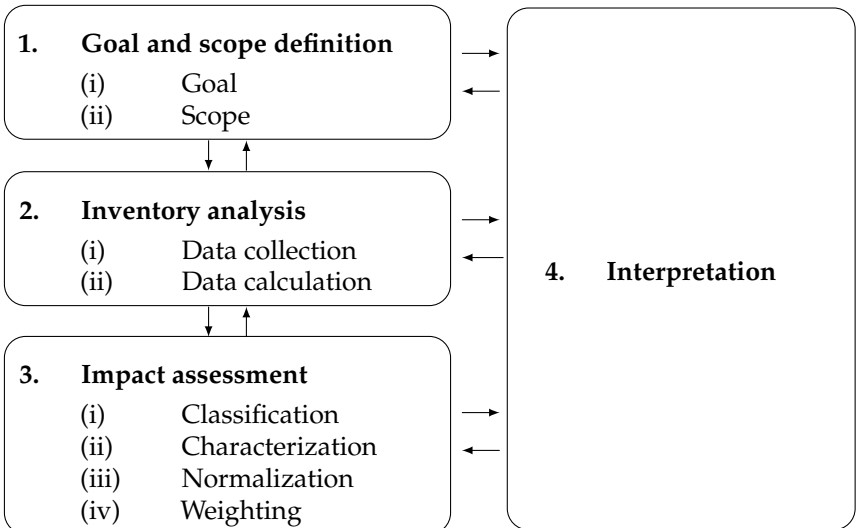

**Figure 1.** Phases of an environmental life cycle assessment, ref. [13], adapted from [3].

In the following, information relevant to the LCA software and its evaluation will be provided. First, additional information on each of the phases shown in Figure 1 is given. Then, a general distinction between two LCA approaches is explained.

The **goal and scope definition** comprises the intended application and the reasons for the implementation. Among others, a unit is established, the so-called *functional unit*, which defines the product's performance in a quantity. Additionally, system boundaries are defined. The life cycle ideally refers to the entire lifetime known as *cradle-to-grave*, but can also be narrowed down to *cradle-to-gate*, *gate-to-gate*, or *gate-to-grave* [3,4].

Inputs and outputs (also referred to as "resources and emissions") that occur during the life cycle of the process or product system are quantified in an **inventory analysis**, also known as life cycle inventory analysis (LCI). An inventory can be set up based on the literature, laboratory test data, industry knowledge from, e.g., the production chain, or electronic databases. Examples of electronic databases are the *European Reference Life Cycle Database (ELCD)* [14] (which has been discontinued as of the 29 June 2018) and the database from *ecoinvent* [15], which is partially freely available at the open access database by *GLAD* [10]. The high level of detail that characterizes these electronic databases needs to be highlighted. In these databases, flows are subdivided into so-called compartments (e.g., emission "to water", "to soil" or "to air") and sub-compartments (e.g., "freshwater" or "sea water"). The resulting impact can vary depending on how the flow is used. To show the effect on the assessment, an example is given at the end of the section in Figure 2.

In the **impact assessment**, also known as life cycle impact assessment (LCIA), the magnitude and significance of the LCI are evaluated based on characterization factors. Different LCIA methodologies, such as *ReCiPe* [16], or *Eco-Indicator 99* [17], use various impact assessment models and assessment factors to quantify the potential impacts [18]. In the following, *ReCiPe* will be explained exemplary (more details in [13]). The *ReCiPe* method is a combined mid- and endpoint category model. *Midpoint categories* represent indicators that measure specific environmental stressors, such as "climate change" (also known as *Global Warming Potential* (GWP)). It quantifies the emissions of greenhouse gases in $gCO_2$-equivalent. Whereas midpoints are closer defined to the emission itself, the *endpoint categories* express the impact as societal damage, such as the three areas of protection of human health, ecosystem quality, and resource scarcity. Endpoints are more comprehensible, e.g., the damage to ecosystems is measured in loss of species in a year ([species.year]). They make LCA results more accessible to decision-makers and the public, but are also more subject to assumptions and uncertainties. To handle these uncertainties, *ReCiPe* offers different perspectives (*individualist*, *egalitarian*, and *hierarchist*) and regions (*Europe* and *world*), which affect the factor of characterization (conversion of in- or outputs to impact indicators),

normalization (e.g., the region's population) and weighting (e.g., the perspective's time frame) within the calculation.

The *ReCiPe* method also provides the option of a *single score*, which represents the impact information in one metric often used to compare the overall environmental performance of different products with each other. This idea originated from the concept of Wagernagel and Rees and the "ecological footprint" [19]. This is an indicator that measures the ecological resources and services required to support a specific human activity or the production of a product. This idea was taken up again in the development of the *Eco-Indicator 99* [17], which resulted in the indicator later known as *single score* in units *Eco-Indicator-Points (pt)*.

To understand how the inventory and its assessment are computed, Figure 2 visualizes the different steps of an imaginary product. It focuses on highlighting the previously mentioned complexity, e.g., the compartments, whereas the mathematical description can be found in [13].

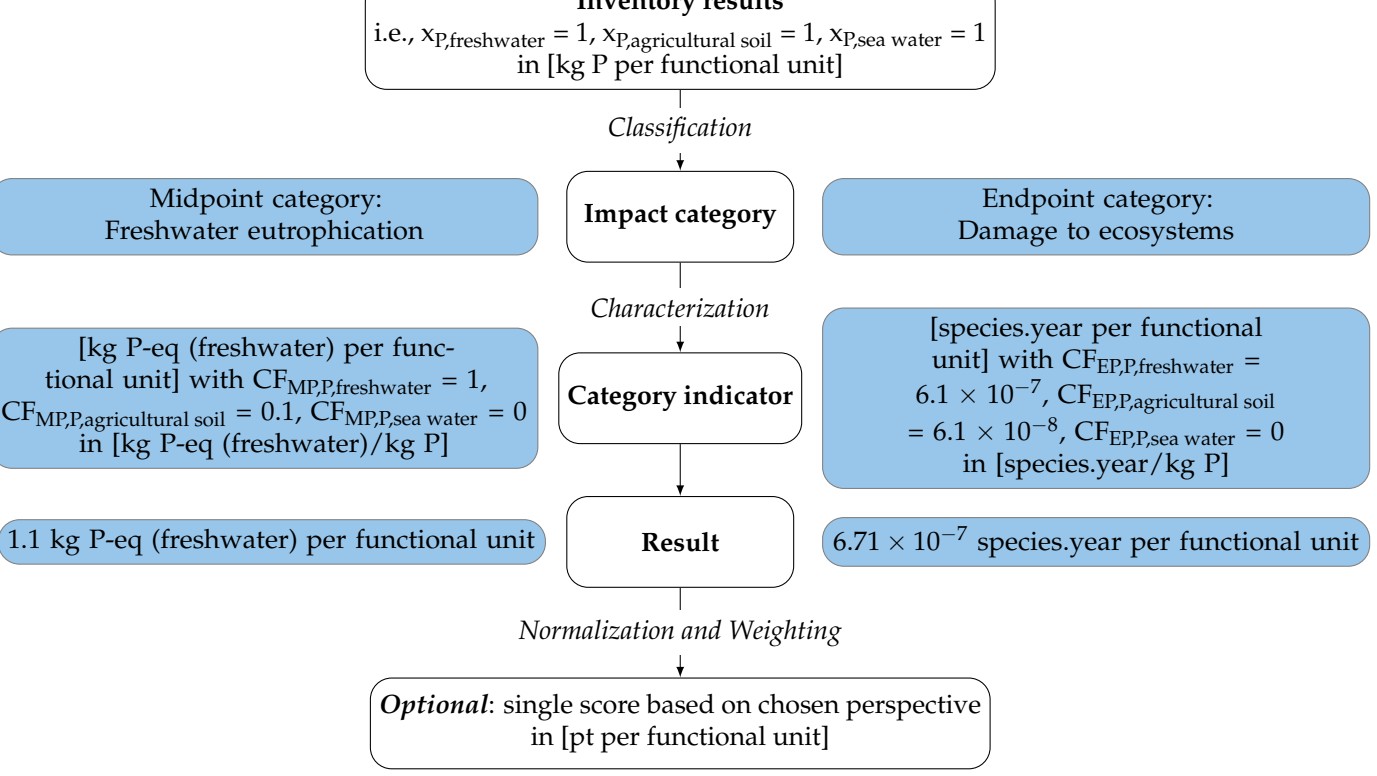

**Figure 2.** Steps during the impact assessment phase on the example of Phosphorus (P), adapted from [13] and [4] (Figure 3 in [4]); $x_{P,freshwater}$ stands for the amount of P emitted in freshwater.

As shown in the figure, it is assumed that in the inventory, 1 kg of phosphorus is emitted into freshwater, 1 kg into agricultural soil, and 1 kg into sea water. These emissions are classified as affecting the freshwater eutrophication (midpoint category) and damaging the ecosystems (endpoint category). To evaluate the emissions, they have to be characterized by indicators: *kg P-eq (freshwater)* for the midpoint and *species.year* for the endpoint. The *characterization factors CF* are used to summarize the emissions in these indicators, e.g., the midpoint CF for emitting to the sea water is 0 and, therefore, assumed to have no environmental impact. If the factors are multiplied by the emissions and then added together, the results are obtained. It is optional to weigh and normalize the results in order to calculate the single score.

As previously mentioned, these steps align with ISO 14040 and 14044 standards. Two distinct approaches are considered for completeness: process-based LCA (traditional ISO) and Economic Input-Output LCA (EIO-LCA). The traditional approach adopts a bottom-up

methodology, dissecting the product's life cycle into individual stages. For example, in the process-LCA of making a sandwich, each step, from ingredient cultivation and transportation to assembly and waste management, is examined, enhancing accuracy but requiring extensive data collection. Conversely, EIO-LCA is a simpler methodology, calculating energy, material resources, and associated greenhouse gas emissions for economic activity. In the sandwich example, EIO-LCA views the product as an economic activity, focusing on economic transactions, such as ingredient purchase, and considering the entire supply chain, encompassing entities like bakeries and wheat farms involved in bread production.

### 2.2. Current LCA Software and Tools

A LCA can be carried out with a variety of software and tools. The most commonly used process-based LCA software and tools are listed and evaluated in Table 1. The following categories used in the table base on Bach and Hildebrand [20]:

- **Origin**, including the country, developer, and year of publication, provides background information and indicates the relevance;
- **Required user knowledge** implies the degree of editable pre-settings for the user, while e.g., for a researcher, a significant degree of flexibility is desired, the goal in teaching is to demonstrate the basic concept;
- **Documentation and tutorials** highlights which tools provide sufficient material to use the software;
- **Accessibility** distinguishes between free, conditional (e.g., free for educational and paid for professional use), and paid access;
- **Data source** shows whether the software is open for imports or has a predefined database;
- **LCIA methods** implies the degree of flexibility with respect to importing different assessment methods;
- **Sensitivity analysis** shows whether uncertainties can be analyzed;
- **API integration and model customization** refers to possible Application Programming Interfaces (API) and manual adaption of models (such as LCIA) to customize the processing of data.

**Table 1.** Comparison of a selection of software and tools for process-based LCA.

| | SimaPro | LCA for Experts Former GaBi | *openLCA* | Brightway2 | LCA-AD |
|---|---|---|---|---|---|
| **Origin** | Netherlands, PRé Consultants, 1990 [21] | Germany, PE International, 1992 [22] | Germany, GreenDelta, 2010 [23] | Switzerland, Paul Scherrer Institut, 2011 [24] | Germany, Technical University of Munich, 2017 (2022 extended) [12,13] |
| **Required user knowledge** | expert | expert | basic/expert | expert | basic |
| **Documentation and tutorials** | available | available | available | available | limited |
| **Accessibility** | paid | paid | free (paid add-ons) | open source | open source |
| **Data sources** | can import external data | can import external data | can import external data | can import external data | Uses ELCD |
| **LCIA methods** | various | various | various | various | ReCiPe only |
| **Sensitivity analysis** | yes | yes | yes | yes | yes |
| **API integration and model customization** | yes (COM-interface) | no | yes (Java API) | yes (Python API) | yes |

Five tools are compared in Table 1. **SimaPro** is a comprehensive LCA software known for its user-friendly interface and extensive database of life cycle inventory data. It offers various impact assessment methods and suits users looking for a versatile and well-supported LCA tool. **LCA for Experts** (formerly GaBi) is a powerful LCA software that provides access to a comprehensive database of life cycle data and is known for robust modeling capabilities. *openLCA* is an open-source LCA software known for its flexibility and customization options. It offers a wide range of impact assessment methods. It is suitable for users who prefer a free and open platform that can adapt and extend its functionality to meet specific project needs. **Brightway2** is a newer flexible open-source LCA software that stands out for its strong emphasis on customization and adaptability. It provides a Python-based platform for users who want to create customized LCA workflows, integrate with external systems, and perform advanced analyses. It is a powerful choice for users with coding skills and specific project requirements. **LCA-AD** is an open-source tool specifically designed to evaluate aircraft designs. Details are given in Section 2.4. Not considered tools within this study are, e.g., *Umberto* [25] and *CMLCA* [26].

Studies show that these tools can generate different results for the same product system, as mentioned in Section 1. Whereas there are, to the author's knowledge, no comparative tool assessments in the aviation context, studies were conducted relating generalized processes, or specific products. Herrmann and Moltesen [5] compared the outputs from 100 unit processes for *SimaPro* and *GaBi*. They concluded that while the results in many cases are the same, in others, they show discrepancies that may affect the interpretation provided by the LCA study. Speck et al. [27] studied *SimaPro* and *GaBi* as well, and assessed four basic material production and disposal processes for three different LCIAs with a total of 42 categories. They reported that half of these categories show at least a ±20% discrepancy for one material. Additionally, the material's ranking (high vs. low environmental impact) shifted for one category. Bach et al. [20,28] compared different LCA software for the building sector but focused on a qualitative assessment. Emami et al. [29] assessed *SimaPro* and *GaBi* for two buildings and supports the before-mentioned findings that the tool choice affects the LCA interpretation. Lopes Silva et al. [6] compared *SimaPro*, *GaBi*, *Umberto*, and *openLCA*. The researchers conducted a comparative study using a standard case study of particleboard production in Brazil. They analyzed the inventory flows, characterized and normalized impact potentials, and compared the results across the different software tools. The findings show that, in general, the impact results were similar for most impact categories across the software tools. However, there were variations in impact values for the photochemical oxidant formation and freshwater ecotoxicity categories. Additionally, the analysis of the characterization factors used by each software tool revealed several differences. Some software tools had missing CFs, additional CFs, or different CFs for the same flows. These discrepancies contributed to the differences observed in the impact results. Miranda Xicotencatl et al. [30] analyzed the effect of LCI database versions and software choice (*Brightway* and *CMLCA*). They tested the hypothesis of whether the same data and modeling yield the same result on the example a of permanent magnet. They found out that the percentage difference between the tools can be below 0.4% if the inventory-LCIA linkage is correctly implemented.

## 2.3. Challenges of LCA in Aviation and Aircraft Design

In addition to the general overview given in Section 2.1, this section focuses on highlighting the challenges of LCA in the aviation sector and the conceptual aircraft design.

The first challenge is the multidisciplinarity. In the systematic literature review of Pinheiro Melo et al. [7], aviation's life cycle is presented based on three systems: (1) the aircraft, (2) the infrastructure, and (3) the fuel life cycle. They all include phases of resource extraction, manufacturing/construction/production, the operation and maintenance of the system, and its end-of-life. Even though these systems could be assessed separately, they are highly connected. Evaluating new technologies might shift, e.g., the relevance of the aircraft's operating phase to the fuel production [13].

Assessing future technologies leads to another difficulty. Due to the wide range of technical parameter variations and temporal and geographical variability, they are characterized by a high degree of uncertainty, e.g., the effects of future aircraft systems may be influenced by regional variations in the electricity mix or the utilized materials. It is possible for a component to be manufactured in one country and have some of its raw materials imported from another [7].

Another challenge is the often inaccessible data. The systematic literature review of Keiser et al. [8] examined, among others, the data used for LCA conducted in the aviation domain and highlighted its limitations [8]. 64% of the authors consult scientific literature, while 47% also use the electronic database of *ecoinvent*.

Finally, the special nature of flying at different altitudes should be mentioned. Current impact assessments characterize emissions based on emissions close to sea level. However, this does not take into account the climate impact that results from flying at different altitudes causing non-$CO_2$ emissions such as $NO_x$, water vapor, and soot, as well as the cloud or contrail formation. In order to consider the non-$CO_2$ effects and altitude dependency, climate models can be integrated into the assessment (e.g., [31]). However, it should be noted that they are subjected to a low degree of confidence [32].

### 2.4. Previous Aircraft LCA Work

To provide a comprehensive overview of LCA in aviation, Keiser et al. [8] conducted a systematic literature review to identify the state-of-the-art and common approaches to aircraft LCA [8]. A total of 45 publications are analyzed qualitatively, covering a wide range of research areas within the aviation industry, including energy-efficient aircraft design, optimizing aircraft ground operations and facilities, sustainable aircraft production, and more (119 are analyzed quantitatively, including the new aircraft fuel development studies, which are already reviewed in [7]). The results show that 23 studies focus on the design of energy-efficient aircraft, especially the research of fiber composite materials (19 studies). Additionally, the following topics are discussed in detail: (1) what aircraft components are studied; (2) which functional units are used; (3) which system boundaries are defined; (4) which data sources, impact assessment methods, and software tools are utilized; (5) what environmental indicator is used; and (6) how the studies document their LCA approach.

Of interest in this study are the functional units, LCIAs, the used software, and environmental indicators. Firstly, the unit "passenger kilometer (PKM)" is, according to the analysis, considered to be appropriate for the entire aircraft operation. Regarding the LCIA, it is to be highlighted that 31% of the studies do not report any impact assessment method. Besides that, the dominant estimation method is *ReCiPe*, used by 27% of the authors, followed by *Eco-Indicator 99* with 11%. Furthermore, minor details about the software or tool being used are provided. In general, it is reported that 64% of the papers use LCA software. At 36%, *SimaPro* is the most commonly used (e.g., [33,34]), followed by *GaBi* (e.g., [35]) and *OpenLCA* (e.g., [36,37]) at 14% each. In total, 20% of the studies use simplified methods, such as EIO-LCA [38] (e.g., [39]) or Greenhouse Gases, Regulated Emissions, and Energy Use in Transportation (GREET) [40]. One study uses its own developed tool (the Eco-Efficiency-Assessment-Model by DLR [41]), and for the remaining studies, no accounting software or tools were provided. Lastly, an analysis is conducted on environmental indicators. In total, 87% correspond to the midpoint categories, whereas 82% relate to the "climate change" category. The "photochemical oxidant formation" (commonly referred to as summer smog), "resource depletion", "acidification", and "human toxicity" are also usually taken into account. In addition, the endpoint categories are also determined by 27% of the analyzed literature. The review makes no note of the single score.

A few additional studies are worth mentioning. One of these studies is from Rahn et al. [42,43]. To eliminate simplification, such as considering flight hours as an average per year or neglecting maintenance events, they suggest a discrete-event simulation combined with LCA [42]. In a more recent publication [43], the authors analyze different dynamization methods to allow detailed consideration of temporal and spatial variations.

An application case is, e.g., the altitude-dependency operating an aircraft, which, therefore, requires dynamic characterization.

Furthermore, Johanning's study [12] should be highlighted, as this tool serves as a baseline within this study. He aimed to build a simple and comprehensive connection between LCA and aircraft design. The *Microsoft Excel* tool *LCA-AD* containing the model is open-source [44], as well as its updated version as *MATLAB* scripts [13]. The tool is limited to up to 13 inputs (e.g., number of seats, operating aircraft empty weight, including default values) assessing aircraft powered by kerosene, sustainable aviation fuels, hydrogen, or battery-electric. The used functional unit is PKM, and the data bases on a mix of literature and *ELCD* data. *ReCiPe* is used as LCIA, which calculates mid- and endpoint categories, as well as the single score. A process-based LCA approach is used including a *cradle-to-grave* system boundary. His assumptions are the following:

- **Goal and scope definition:** environmental impact of a short-range passenger aircraft (A320-200 like), *openLCA 1.4.1* as LCA software with following processes:
  - **Design and development:** computer usage, wind tunnel tests, flight test campaign;
  - **Production:** use of production facilities, material production;
  - **Operation:** energy generation and consumption at airports, ground handling at airports, and either:
    * For kerosene, sustainable aviation fuel, or hydrogen-powered aircraft: fuel production, cruise flight, landing, and take-off cycle;
    * For battery-powered aircraft: battery production, battery charging;
  - **End-of-life:** reuse.
- **Inventory analysis:** *ELCD 3.0* database;
- **Impact assessment:** *ReCiPe 2008* (was updated to *ReCiPe 2016* with [13]) & altitude-dependent linear climate model;
- **Interpretation:** based on the *single score* (assumptions: hierarchist perspective with average weighting and region world).

The methodology is outlined in Figure 3.

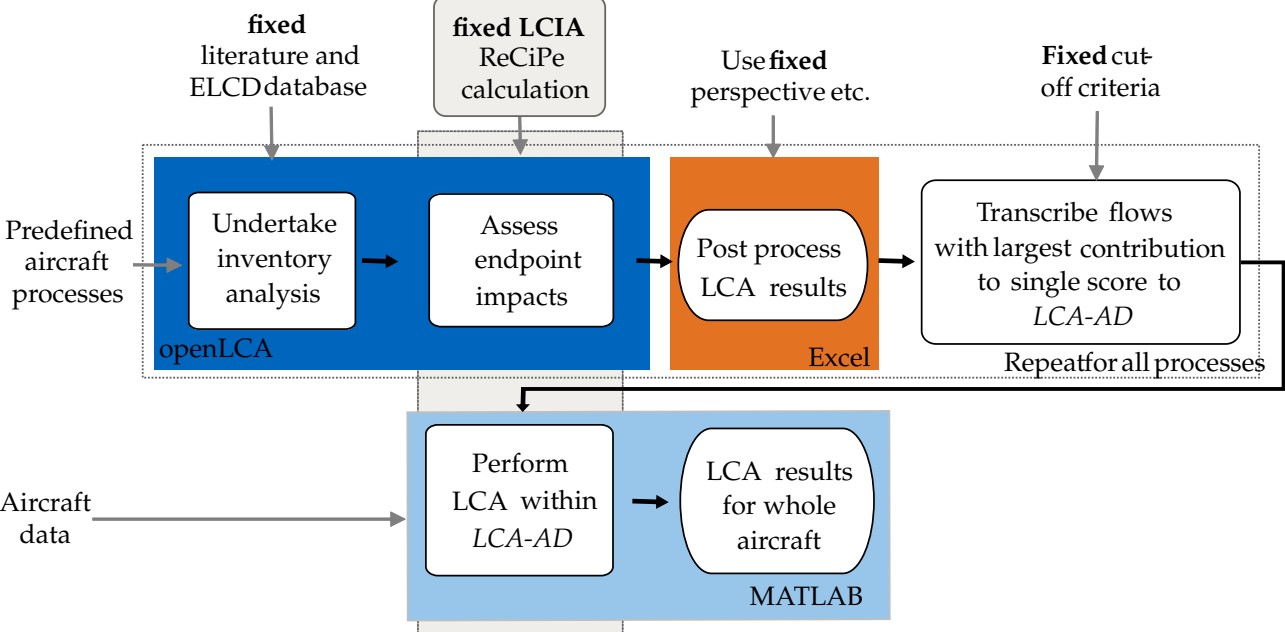

**Figure 3.** Methodology of simplified *LCA-AD* tool of Johanning [12], adapted from [13].

Starting with a fixed database, LCIA, perspective (hierarchist), and cut-off criteria, an analysis is conducted for each process in *openLCA*, e.g., for the process "computer usage", a LCI is conducted, which results in a list of resources and emissions (can be up to

800 in- and outputs). These are evaluated then with the *ReCiPe* endpoint assessment, and the results are exported to an *Excel* spreadsheet. To reduce the extensive list, in- and outputs that fall below a fixed threshold of single score contribution are not considered relevant (up to 2.5%). Additionally, to reduce the complexity, the differentiation between subcompartments (freshwater, ocean, etc.) is neglected by using the emission and resource with the highest *CF*. In the case of "computer usage", the inventory is reduced from 392 to 13 in- and outputs (1% threshold), which cover 94.2% of the overall modeled process. The reduced LCI is then transcribed to the *LCA-AD* tool, where the overall aircraft assessment is conducted.

This approach is easy to use but reduces the share of considered in- and outputs, limits the tool in the assumed settings, and does not allow, e.g., a new perspective, weighting, region, cut-offs, database, or impact assessment method to be used.

## 3. LCA Tool Evaluation Methodology

### 3.1. Assumptions and Limitations

The basis of the study is the short-range aircraft Airbus A320-200, including the same processes as described in Section 2.4 (the configuration is described in detail in [12,13] and in Appendix A). Five different data sets were studied: an aircraft powered by (1) kerosene, (2) hydrogen based on steam methane reforming (SMR), (3) hydrogen produced from electrolysis, (4) a battery with electricity based on the EU mix, or (5) powered by a battery with electricity from renewable energies (renew.). It should be emphasized that the battery-electric configuration has half the range of the other two concepts. The altitude-dependent climate model from Dahlmann [31] is included (the switch from [45,46] is outlined in [47]).

To differentiate between the models, *LCA-AD* is synonymously called the *simplified tool*. To be mentioned is that minor bugs were noticed during its use, which were fixed (see Appendix B.1). The implemented new interface outlined in Section 4 is called *openLCA-AD* or the *advanced tool*. The following prerequisites and limitations are given:

- Installation of *Python* 3.X and libraries (*olca-ipc* (0.0.12) and *pathlib* (1.0.1));
- Installation and configuration of *openLCA* software (least 1.11.0; will also include *Java*);
- Creation of database including the respective product system "aircraft" (with all modeled processes) and import of LCIAs in *openLCA*;
- Limitation to Windows only, and might differ for Linux and OS X.

### 3.2. Quantitative and Qualitative Metrics

This section aims to answer the research question **RQ1**. The *quantitative metrics* include the numeric values to be assessed, whereas the *qualitative metrics* refer to soft indicators, which relate to the software handling. An overview is given in Table 2.

It was decided that the *quantitative metrics*, the *single score* and the endpoint categories "damage to human health", "damage to ecosystems", and "damage to resource availability" are prioritized as high, despite the fact that they are not or only mentioned in 27% of the analyzed studies of [8]. Because of the mentioned uncertainties for aircraft LCA results (Section 2.3), it is preferable to consider the results as an overall decision-making indicator (main purpose of endpoints) or as relative impacts of new concepts compared to a reference concept (main purpose of *single score*).

Additionally, the midpoint "climate change" is the highest-ranked quantitative metric in the review of [8] and this study. Furthermore, low-priority qualitative metrics are midpoints considered up to 40% and 40–80% as mid-priority.

The *qualitative metrics* are conducted from an own analysis when using LCA tools. It should therefore be emphasized that they are based on a subjective evaluation and prioritization. The ability to trace the results (*diagnostic use*) and to perform sensitivity studies (*uncertainty analysis*) is ranked as a high priority. As a researcher, it is considered important to be able to trace and understand the outcomes of their analyses, ensuring transparency and reliability. Additionally, assessing the impact of uncertainties on the results is considered important, enhancing the software's overall credibility.

Mid-prioritized are the abilities to adapt the database and assessment methods (*LCI and LCIA adaptability*), the vulnerability to errors (*implementation stability*), and the modeling's accuracy (*level-of-detail*). The ability of database and method adaptability is an important feature, whereby a certain amount of effort (familiarization with the system, validation of the interface/import) is considered acceptable. The importance of a stable and error-resistant software environment is acknowledged and considered as a topic of a robust software architecture design. This can reduce the modeling and validation process. However, the underlying, more critical goal of ensuring reliable data are covered in the metric of *diagnostic use*. Model accuracy, represented by the level-of-detail, is deemed important but not the highest priority, considering the overall objective of conceptual aircraft design.

**Table 2.** Qualitative and quantitative metrics for LCA tool evaluation in conceptual aircraft design.

| | Metric | Priority | Comment * |
|---|---|---|---|
| **Quantitative** | Single score | high | not mentioned |
| | Damage to human health | high | 27% |
| | Damage to ecosystems | high | 27% |
| | Damage to resource availability | high | 27% |
| | Climate Change | high | 82% |
| | (Photochemical oxidant formation) summer smog | mid | 44% |
| | (Mineral and fossil) resource scarcity | mid | 44% |
| | Terrestrial acidification | mid | 44% |
| | Human (non- and carcinogenic) toxicity | mid | 42% |
| | Particulate matter formation | low | 38% |
| | (Freshwater and marine) eutrophication | low | 33% |
| | (Terrestrial, freshwater, marine) ecotoxicity | low | 31% |
| | Ozone depletion | low | 29% |
| | (Land) natural area use | low | 24% |
| | Ionizing radiation | low | 22% |
| | Water consumption | low | 11% |
| **Qualitative** | Diagnostic use | high | How efficient is the software in terms of result traceability? |
| | Uncertainty analysis | high | To what degree are uncertainty analysis possible? |
| | LCI and LCIA adaptability | mid | What is the degree of difficulty involved in extending the software with respect to new databases or assessment methods? |
| | Implementation stability | mid | To what extent does the software's implementation exhibit vulnerability to errors? |
| | Level-of-detail | mid | What is the depth of detail included in the software's modeling? |
| | User-friendliness | low | To what extent does the software exhibit ease of use? |
| | Time effort | low | To what extent does the modeling require time investments? |

* percentage values refer to [8].

*User-friendliness* and *time effort* were ranked as low priority. While user-friendliness is undoubtedly important, the assumption here is that the users will have the necessary expertise to prioritize functionality over ease of use. Similarly, time effort is assigned a lower priority, considering that time may be sacrificed to some extent to achieve, e.g., traceable results.

## 4. Implementation of *openLCA-AD*

To overcome the shortages of the simplified *LCA-AD* tool mentioned in Section 2.4, a direct connection to the *openLCA* software is possible. The in-house aircraft design environment Aircraft Design Box (*ADEBO*) of the Chair of Aircraft Design at the Technical University of Munich is implemented in the coding language *MATLAB*, which communicates with *openLCA* over an Application Programming Interface (API), the programming language *Python* and its package *olca-ipc*. The prerequisites are mentioned in Section 3.1. Further documentation and examples can be found in [48].

The methodology of the advanced tool *openLCA-AD* is outlined in Figure 4, showing that the overall assessment is performed in *openLCA*. For that, inputs are set firstly in

*MATLAB*, such as the selected assessment method and assumptions (e.g., ReCiPe (2016) hierarchist and region world), the aircraft-specific data (e.g., aircraft's maximum take-off weight), as well as additional data pre-processing (e.g., estimating turn-around time based on aircraft type). With the input data, a *Python* script is called, which includes the *olca-ipc* package [49] providing an API for inter-process communication with *openLCA*. With that, all modeled processes (from the development to the reuse phase) can be calculated at once with different LCIA and perspectives. After the simulation exports the results, the data are post-processed. The quantitative metrics presented in Section 3 are either read in (e.g., results of midpoint and endpoint categories) or require additional calculation (e.g., assessing specific LCI).

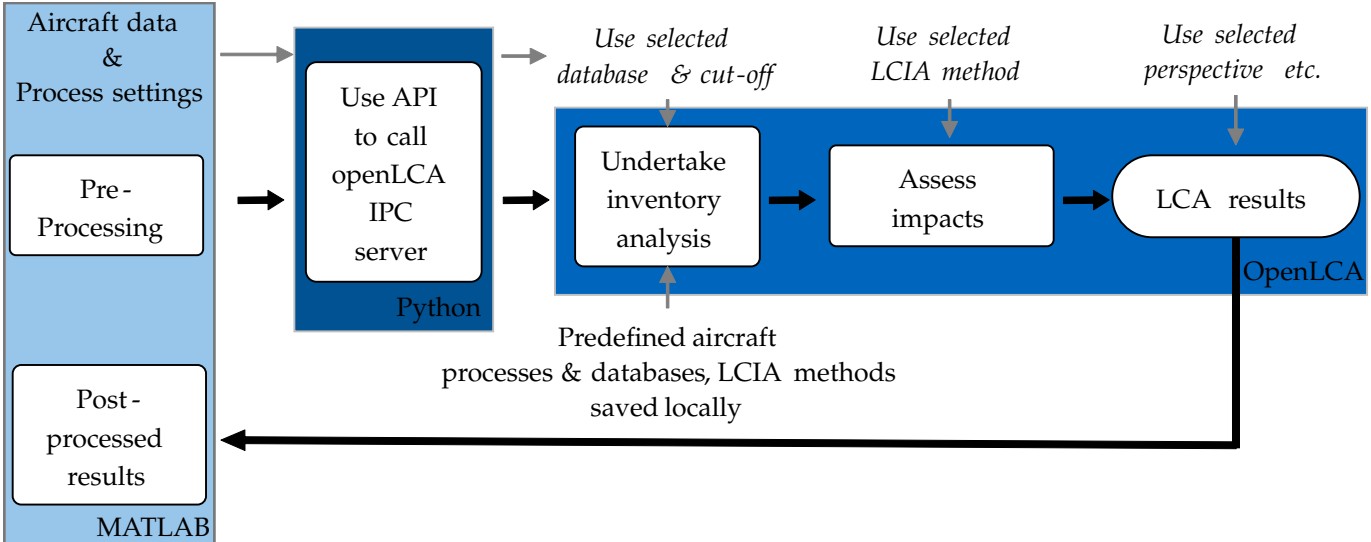

**Figure 4.** Methodology of advanced *openLCA-AD* tool (automated interface *MATLAB-Python-openLCA-MATLAB*).

## 5. Results and Discussion

### 5.1. Quantitative Metrics

This section aims to answer the research question **RQ2**. The results refer to the *simplified LCA-AD* tool of [12,13] and the *advanced openLCA-AD* tool outlined in Section 4. For a fair comparison, both refer to the same LCIA method (*ReCiPe 2016*), perspective (hierarchist), weighting (average), region (world), and include the same altitude-dependent climate model. The bug fixes of *LCA-AD* (see Section 3.1) have minor impact on the results and are therefore not discussed in detail. An overview is given in Table A3.

The selected metrics are explained in Section 3.2. First, the high-priority metrics *single score*, the endpoint categories, and the midpoint "climate change" are discussed in detail. A broader review of the mid- and low-priority metrics is given afterward. Even if the life cycle inventory is not addressed in depth, it is the foundation of the majority of the metrics' differences. Therefore, details exemplary on the kerosene-powered aircraft are available in Appendix B.2 and Table A4. Additionally, Tables A5–A8 provide the assessments of all concepts.

#### 5.1.1. High-Priority Metrics

The initial metric under consideration is the *single score* (SS), as illustrated in Figure 5. This figure displays the relative differences in SS between the kerosene-powered aircraft and its alternatives using both the *simplified* and *advanced tools*. The findings indicate that the methodologies exert a negligible difference on the single score. The SS impact ranges from $-1.2\%$ for the hydrogen-powered aircraft concept with $H_2$ generated by electrolysis, to $+8\%$ for the same concept with hydrogen generated by steam methane reforming. The reason lies, for one, in the reduced number of considered in- and outputs of the *simplified tool* as

explained in Section 2.4. For example, the input "hard coal" was neglected in the process "energy carrier production" which was considered per 1 MJ energy or kg fuel. Multiplying the resource with the actual energy amount, the considered resource increases by a factor of $10^3$ (see also reduced numbers in Table A4 and detailed explanation in the Appendix B.2). For another, in- and outputs are neglected in the *simplified tool* if the characterization factor of the sub-compartment is small. For example, for the emission "phosphate" ($PO_4$), only "emission to air" (and not "to soil") is considered. Overall, this leads to an increased SS for all configurations assessed by the *advanced tool* when comparing it with the *simplified* one (see Tables A5–A8).

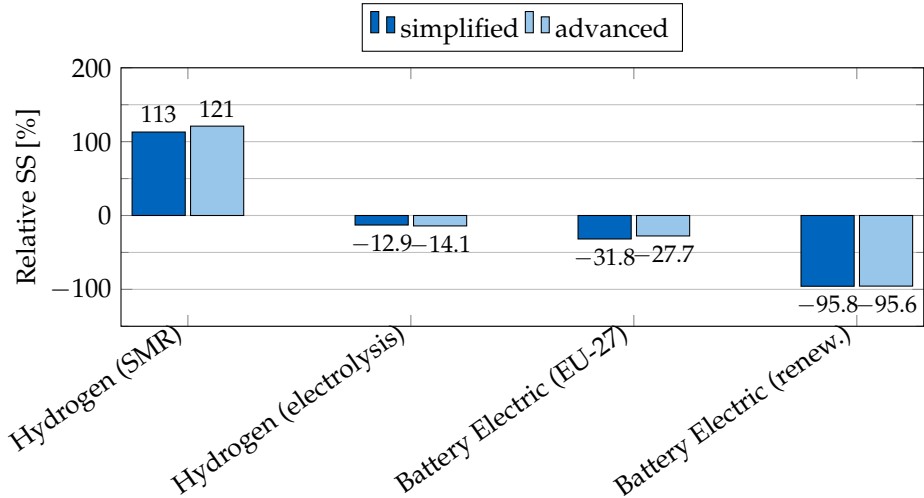

**Figure 5.** Single score comparison for *simplified LCA-AD* vs. *advanced openLCA-AD* tools for individual fuels relative to kerosene-powered aircraft.

A similar trend is evident in Figure 6, portraying the *single score* shares of the modeled processes in comparison of the simplified and advanced tool and all concepts (processes are listed in Section 2.4).

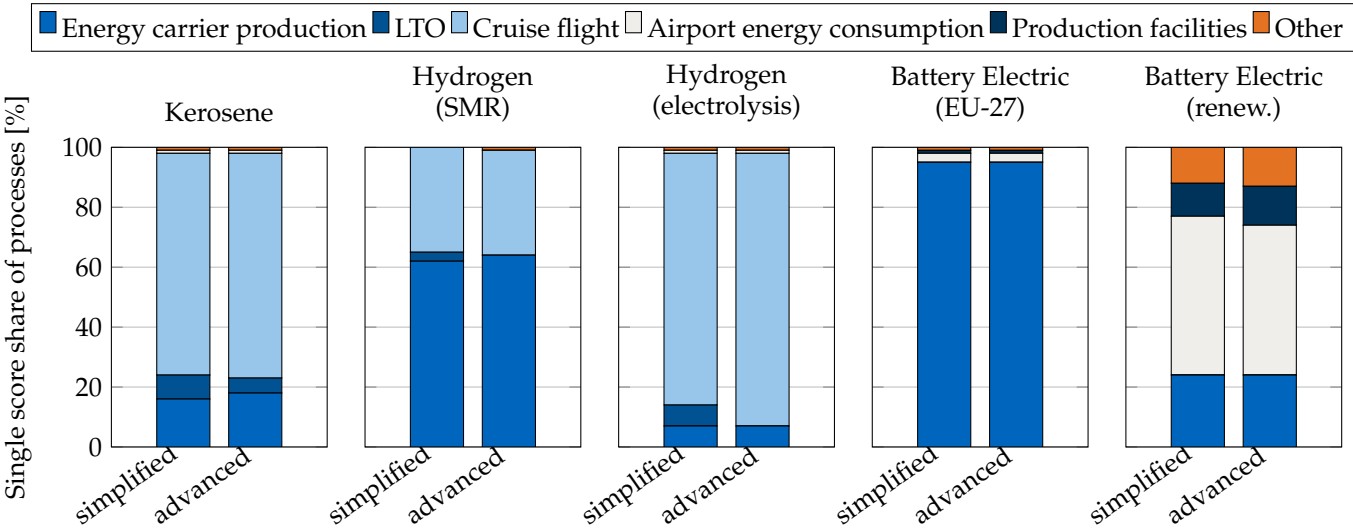

**Figure 6.** Comparison of the single score shares of life cycle processes on different aircraft concepts for the *simplified LCA-AD* vs. *advanced openLCA-AD* tools.

For most aircraft, the processes "energy carrier production" (kerosene, liquid hydrogen, or electricity production for battery charging, respectively), "LTO" (landing and take-off), "cruise flight", "production facilities", and "airport consumption" have a predominant

impact. The "other" category aggregates the remaining processes. The results reveal that the tools show a marginal difference in the single score shares. In the case of kerosene-powered aircraft, the share for "energy carrier production" slightly increases from 16 to 18%, "LTO" decreases from 8 to 5%, and "cruise flight" increases from 74 to 75%. For both hydrogen-powered concepts, "LTO" shares reduce by less than 1% and shift to an increased "energy carrier" by 2% for SMR and to "cruise flight" by 7% for the electrolysis version. No changes are observed for the battery-electric aircraft (EU-27). For the battery-electric aircraft (renew.), a slight shift from "airport energy consumption" to "production facilities" is evident (simplified: 53 and 11%, advanced: 50 and 13%).

The subsequent metrics are the *endpoint categories*, depicted in Figure 7. These categories include "damage to human health" in Disability-adjusted life years or DALY per PKM, "damage to ecosystems" in species.year per PKM, and "damage to resource availability" in USD (2013) per PKM presented for all configurations and both *LCA-AD* and *openLCA-AD*.

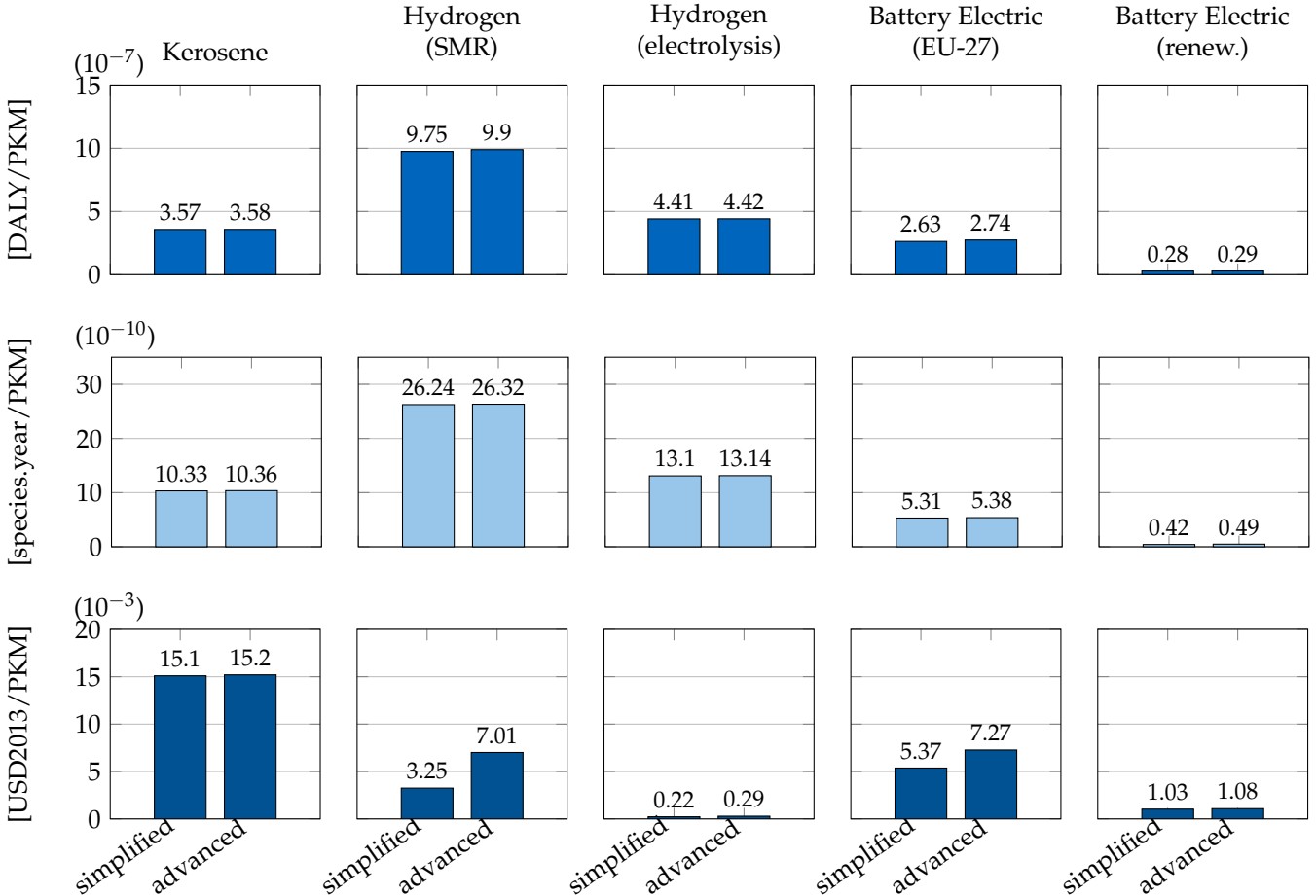

**Figure 7.** Comparison of the endpoint categories "damage to human health" (row 1), "damage to ecosystems" (row 2) and "damage of resource availability" (row 3) for different aircraft concepts for the *simplified LCA-AD* vs. *advanced openLCA-AD* tools.

The outcomes indicate that the *advanced* tool has varied impacts on the endpoints. For the first two categories, minor absolute divergences range between 0.01 and $0.23 \times 10^{-7}$ DALY/PKM and 0.03 and $0.08 \times 10^{-10}$ species.year/PKM. In relative terms, the maximum deviation is observed for "damage to human health" in the battery-electric aircraft (EU-27) with 4.34%, and for "damage to ecosystems" in battery-electric aircraft (renew.) with 15.0%. The category "damage to resource availability" exhibits more substantial differences, ranging from 0.05 to $3.76 \times 10^{-3}$ USD2013/PKM, leading to a maximum relative deviation of 116% for hydrogen-powered aircraft (SMR). This is linked to the

above-mentioned missing "hard coal" resource in the "energy carrier production", which is considered for the current EU electricity mix.

The final qualitative metric under consideration is the *midpoint category "climate change"*, detailed in Figure 8. They are presented for all five concepts using the *simplified* and *advanced tool*. The findings indicate a minor impact of the *openLCA-AD* on "climate change". The midpoint increases by up to 0.99 gCO$_2$ eq/PKM for the hydrogen-powered aircraft (SMR), with the highest relative difference of 2.32% observed for the battery-electric aircraft (renew.).

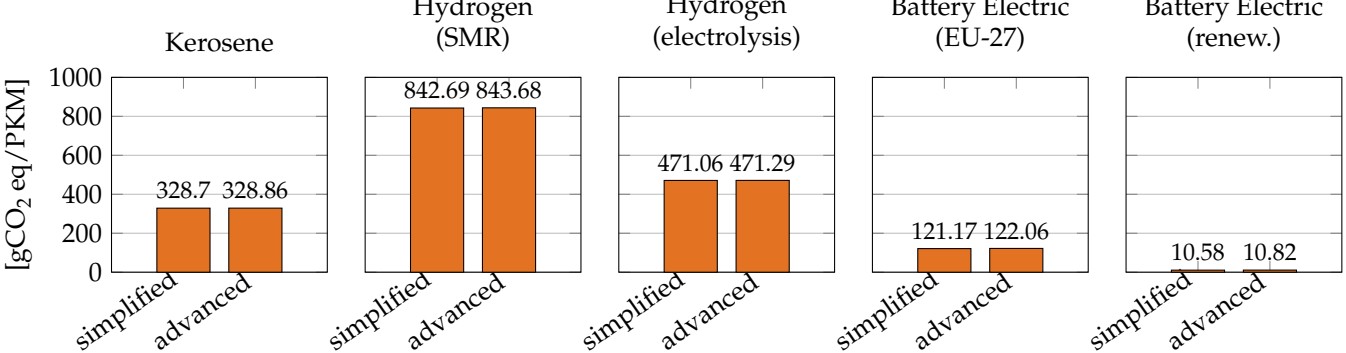

**Figure 8.** Comparison of the midpoint "climate change" for different aircraft concepts for the *simplified LCA-AD* vs. *advanced openLCA-AD* tools.

### 5.1.2. Mid- and Low-Priority Metrics

The mid- and low-priority metrics delve into midpoint categories that are often not considered relevant in aviation research, as outlined in Section 3.2. Consequently, a concise description is given to these categories, and for detailed data it is referred to Tables A5–A8.

Opposed to the low-impacted *single score*, most midpoints are strongly influenced by the tool choice. Among the four mid-priority midpoint categories, "terrestrial acidification" and "summer smog" display disparities of less than 10% between the methodologies. In contrast, "human toxicity" and "resource scarcity" show large variations, e.g., of up to a factor of 80 for "non-carcinogenic toxicity" for the hydrogen-powered aircraft (SMR). This is linked to underestimated arsenic, lead, and zinc per 1 kW of energy from the current electric mix.

Across the seven low-priority midpoint categories, only "particulate matter formation" exhibits differences lower than 10%. The most significant deviations are evident in "ozone depletion," where a two- to four-order magnitude shift occurs for all configurations. This is due to the underestimated presence of dinitrogen monoxide (N$_2$O) (exemplary shown for kerosene-powered aircraft in Table A4).

### 5.2. Qualitative Metrics

To compare the *simplified LCA-AD* and *advanced openLCA-AD* qualitatively, metrics have been established in Section 3.2. Table 3 evaluates them in a decision matrix.

The *simplified* and *advanced tool* are ranked to fulfill 82% and 79% of the indicators, respectively. For the *LCA-AD*, e.g., the **diagnostic use** is ranked as easy to trace and debug due to its modeling in a programming script. The *openLCA-AD*, in contrast, offers visualization of the results, but the calculation steps are not easy to follow and rather non-transparent. The **uncertainty analysis** is ranked high for both, as *openLCA* offers a built-in Monte-Carlo-Simulation, which can be triggered with the API, or script-based parameter variation can be used. In terms of **LCI and LCIA adaptability**, the *simplified tool* does provide poor capabilities (see Figure 3), whereas flexibility for the inventory and assessment methods is high for the interface version. On the other hand, the **implementation stability** for the fully script-based tool is strong. With the help of version control systems like Git, code changes can easily be tracked. The *openLCA* interface can only be tracked for the pre-

and post-processing and the software offers timestamps documenting the last changes. In the *level-of-detail*, a large discrepancy is visible. The ranking is based on these subdivisions:

- LCI (1) with or (2) without cut-offs;
- LCI (1) without or (2) with compartments/subcompartments;
- (1) Without or (2) with spatial/temporal differentiated LCI;
- (1) Only relevant or (2) all processes considered.

The *simplified LCA-AD* considers a low level of detail as it follows the first option for all subdivisions. The *advanced openLCA-AD*, on the other hand, follows the second option for the first two.

The script-based tool beats out the interface in respect of **user-friendliness** and **time effort**. The *advanced tool* does need *openLCA* and *Python* to be installed and requires expert knowledge in the LCA software. The *simplified tool* is a straightforward function requiring a low amount of inputs to generate the results.

**Table 3.** Qualitative metric evaluation of the *simplified LCA-AD* and *advanced openLCA-AD* tool.

| | | Simplified | | Advanced | |
|---|---|---|---|---|---|
| Requirement | Weighting * | Rating ** | Total | Rating ** | Total |
| Diagnostic use | 3 | 3 | 9 | 2 | 6 |
| Uncertainty analysis | 3 | 3 | 9 | 3 | 9 |
| LCI & LCIA adaptability | 2 | 1 | 2 | 3 | 6 |
| Implementation stability | 2 | 3 | 6 | 2 | 4 |
| Level-of-detail | 2 | 0 | 0 | 2 | 4 |
| User-friendliness | 1 | 3 | 3 | 1 | 1 |
| Time effort | 1 | 3 | 3 | 1 | 1 |
| **Points (of maximum 39)** | | | **32** | | **31** |

* Priority low = 1, mid = 2, high = 3. ** Rating from 0 to 3 with 0 as, e.g., for "diagnostic use" 0 for not traceable and 3 for easy to trace.

*5.3. Discussion*

The quantitative metrics assessed in this study reveal varying effects between the *simplified LCA-AD* and the *advanced openLCA-AD* tool on various levels of the LCA. The findings suggest that the use of the different applications has a negligible influence on the overall *single score*. A closer examination of the *single score* shares of the modeled processes supports this observation, as they also experience a marginal impact. However, a substantial impact is demonstrated in the analysis of the endpoint categories. Furthermore, while the impact on the high-priority midpoint category "climate change" is minor, it significantly influences most other midpoints. As the literature lacks validation data, only the midpoint "climate change" of kerosene-powered aircraft can be compared. In Kossarev et al. [13], a comparison of the global warming potentials of kerosene-powered aircraft from different sources is shown. An average GWP for narrow-body aircraft of 118 $gCO_2$-eq/PKM is visualized, which does not consider altitude-dependent climate models. If the model is also excluded in the *simplified* and the *advanced* tool, results of 102.61 and 102.78 $gCO_2$ eq/PKM are achieved, resulting in an underestimation compared to the average values.

In addition to quantitative assessments, qualitative metrics play an important role in evaluating the effectiveness of the methodologies. Both the *simplified* and *advanced* tools are ranked to fulfill >75% of the requirements, but they focus on different use cases.

The *openLCA-AD* tool is recommended for detailed research and comprehensive assessments. It outperforms the simplified tool in a higher level-of-detail and accuracy, due to its direct interface to the LCA software *openLCA*, making it the preferred choice for examining endpoints and the overall midpoint categories. However, drawbacks in terms of medium-level traceability, required expert knowledge, and high modeling effort can lead to an increase in required assessment time. This is not beneficial in the conceptual aircraft design because new technologies should be quickly evaluated.

The *LCA-AD* tool serves as a low-level, user-friendly tool that can provide reasonably accurate results, particularly for comparing single scores across different aircraft types. It proves valuable for indicating ecological tendencies and identifying relevant processes, e.g., in first assessment within the conceptual aircraft design or for teaching. However, its accuracy diminishes when examining endpoints or other midpoints, emphasizing its limitations for in-depth analyses. Additionally, the tool relies on data derived from a specific LCA software. The dependence on a particular software introduces challenges, particularly when new databases become available (see *GLAD*) or when life cycle impact assessments undergo updates. Updating the simplified tool under such circumstances requires a substantial effort.

However, the advent of new automation technologies opens avenues for addressing this limitation. Future work could explore a procedure that automatically creates semi-empirical or surrogate models from LCA software. This approach could streamline the adaptation of the simplified tool to changes in databases or impact assessment methodologies, enhancing its flexibility and ensuring an up-to-date version when used in LCA training and teaching. Additionally, this can increase the transparency of *openLCA-AD* by transcribing relevant inventory data, characterization factors, and results in an easy-to-understand code.

## 6. Conclusions

While several life cycle assessments in the aviation sector have been undertaken, their comparability is hindered by disparities in data sources, tools, and methodologies employed. The current research is increasingly oriented towards establishing a global database and standardized impact methodologies. By providing a comparative analysis of life cycle assessment tools in the context of conceptual aircraft design, this study adds to the ongoing effort.

The analysis includes two research questions: To methodically address the first question, a LCA tool evaluation including quantitative and qualitative metrics is formulated. The quantitative domain encompasses five high-priority indicators, including the single score, three endpoint categories, and the "climate change" midpoint category, alongside eleven mid- or low-priority midpoint categories. Concurrently, the qualitative metrics consist of seven indicators, prioritizing "diagnostic use" and "uncertainty analysis."

The second research question involves the application of these metrics to assess two LCA tools: a simplified open-source tool *LCA-AD* derived from the *openLCA* software, and an automated interface tool *openLCA-AD* developed in this study for with same software. Quantitative metrics reveal variable effects: the tool selection exerts negligible influence on the overall single score and "climate change" midpoint category, yet significantly impacts endpoint and other midpoint categories. Qualitative analysis indicates that both tools satisfy over 75% of the metrics, albeit with preferences for distinct use cases.

The implications of the findings suggest that researchers and practitioners should carefully align their choice of tools with the intended application. The simplified tool is suitable for didactic purposes and quick assessments on single score level. In contrast, the advanced tool is essential for research requiring high levels of LCA expertise, modeling, and time effort to accurately assess the overall impact categories.

In summary, the evaluation methodology presented in this study significantly enhances our grasp of the intricacies involved in comparing LCA tools. It can serve as a guideline that points out difficulties and peculiarities. Applying this method to two tools highlights the need to choose LCA tools carefully for accurate and reliable analyses in conceptual aircraft design.

**Author Contributions:** Conceptualization, K.M.; methodology, K.M. and M.S.; software, K.M. and M.S.; validation, K.M. and M.S.; formal analysis, K.M. and M.S.; investigation, K.M. and M.S.; resources, K.M. and M.S.; Writing—Original Draft Preparation, K.M.; Writing—Review & Editing, M.S. and M.H.; visualization, K.M.; supervision, M.H. All authors have read and agreed to the published version of the manuscript.

**Funding:** This research received no external funding.

**Data Availability Statement:** Data are contained within the article.

**Conflicts of Interest:** The authors declare no conflicts of interest.

## Abbreviations

The following abbreviations are used in this manuscript:

| | |
|---|---|
| ADEBO | Aircraft Design Box |
| API | Application Programming Interface |
| CF | Characterization factor |
| DALY | Disability-adjusted life years |
| EIO-LCA | Economic Input-Output LCA |
| ELCD | European Reference Life Cycle Database |
| EU mix | European electricity mix |
| LCA | Life cycle assessment |
| GLAM | Global Guidance on Environmental Life Cycle Impact Assessment Indicators |
| GLAD | Global LCA Data Access |
| GWP | Global Warming Potential |
| LCA-AD | Life Cycle Assessment in Conceptual Aircraft Design |
| LCI | Life cycle inventory analysis |
| LCIA | Life cycle impact assessment |
| openLCA-AD | *openLCA* in Conceptual Aircraft Design |
| PKM | Passenger kilometer |
| pt | Eco-Indicator-Points |
| renew. | Renewable energies |
| SMR | Steam methane reforming |

## Appendix A. Aircraft Configurations

The reference aircraft is considered as the A320-200 WV000 and equipped with two CFM56-5A5 engines [50]. Two alternative designs, one hydrogen and one battery-electric powered aircraft, were derived from this conventional design by [12]. These aircraft parameters are presented in Table A1.

**Table A1.** Aircraft parameters [12].

| Parameter | Conventional Aircraft | Hydrogen Aircraft | Battery Aircraft |
|---|:---:|:---:|:---:|
| Maximum take-off mass | 73,500 kg | 74,000 kg | 101,300 kg |
| Operating empty mass | 41,244 kg | 48,700 kg | 58,700 kg |
| Trip fuel mass/Energy | 4557 kg | 2790 kg | 15,582 kWh |
| Average passengers per flight | | 122 | |
| Number of operation years | | 25 | |
| Mission range | 589 NM | | 294 NM |

The shares for each material for the three configurations are reported in Table A2. For the hydrogen aircraft, an aluminum tank is considered.

**Table A2.** Shares of materials in aircraft configuration [12,13].

| Material | Aluminum | Steel | Composites | Titan | Other |
|---|:---:|:---:|:---:|:---:|:---:|
| Conventional & electric aircraft [%] | 66 | 9 | 13 | 7 | 5 |
| Hydrogen aircraft [%] | 72 | 7 | 11 | 6 | 4 |

## Appendix B. Results

*Appendix B.1. Single Score*

As errors were noticed in the *simplified tool*, the following is adapted: the characterization factor of *Hydrogen-3* is corrected affecting the midpoint *ionizing radiation* (former referred to the compartment "emission to air" with $CF = 8.56 \times 10^{-4}$ whereas the LCI referred to "emission to water" $CF = 4.12 \times 10^{-5}$). Additionally, for the process "production facilities", the amount of total annual water consumption was given in tons in the previous work, whereas the original source provides the value in cubic meters.

**Table A3.** Comparison of the single score in [pts/PKM] of the aircraft concepts studied in [12,13]. *Simplified* refers to the baseline data in [13], *simplified (fixed)* to the bug-fixed version, and *advanced* to the results from the new interface.

| | **Kerosene** | **Hydrogen** [#] | | **Battery** | |
| --- | --- | --- | --- | --- | --- |
| | | **SMR** | **Electrolysis** | **EU Mix** | **Renew.** |
| **(A) Simplified** [13] | $8.408 \times 10^{-2}$ | $1.786 \times 10^{-1}$ | $7.330 \times 10^{-2}$ | $5.762 \times 10^{-2}$ | $3.963 \times 10^{-3}$ |
| **(B) Simplified (fixed)** | $8.387 \times 10^{-2}$ | $1.784 \times 10^{-1}$ | $7.309 \times 10^{-2}$ | $5.721 \times 10^{-2}$ | $3.551 \times 10^{-3}$ |
| **(C) Advanced** | $8.554 \times 10^{-2}$ | $1.887 \times 10^{-2}$ | $7.349 \times 10^{-2}$ | $6.188 \times 10^{-2}$ | $3.768 \times 10^{-3}$ |
| **Difference** $[(C - B)/B]$ | +1.99% | +5.77% | +0.55% | +8.16% | +6.12% |

[#] New studies show that the Contrail and Cirrus cloud formation is reduced for hydrogen in comparison to kerosene burning concepts (Section 2.2.2 in [51]). For a 40% reduction, the SS in the *simplified* and *advanced LCA* would be $1.536 \times 10^{-1}$ and $1.637 \times 10^{-1}$ for the SMR and $4.836 \times 10^{-2}$ and $4.855 \times 10^{-2}$ for the electrolysis. To focus on the methodology comparison with [13], no reduction was chosen for the above-mentioned results.

*Appendix B.2. Inventory Analysis*

The inventory analysis for the conventional aircraft is reported in Table A4. Only the results of the respective substances previously presented in [13] are provided, even though the automated interface can include all substances for all subcompartments without a cut-off criteria. The inventory analysis is reported for a conventional aircraft only to highlight the general difficulties in inventory assessment. To understand the occurring differences, first, general notes about the differences in how the two methodologies are handling the inventory are given. *Simplified* evaluates the inventory **process-by-process**. Each process was calculated in *openLCA* evaluating one item of its own functional unit (e.g., "kerosene production" for 1 kg fuel, "computer usage" for 1 kW h, "aircraft production" for 1 seat). The single score for this process is estimated and the flow's contribution to that SS is calculated (e.g., the resource use of "crude oil" [g/1 kg fuel] has a SS share of 67.51% of the process "kerosene production). All flows with a greater impact than 1% are assumed to be relevant. In an additional step, the flow is converted in the unit [g/PKM].

*Advanced* evaluates the inventory **in one process**. The processes already include the overall functional unit of PKM, meaning that, e.g., the process "kerosene production" includes the total fuel mass and the aircraft production the total number of seats. The relevant contribution of a flow is now referred to the total SS in [g/PKM].

**Table A4.** Results of life cycle inventory (simplified [13] vs. advanced LCA) of the kerosene-powered concept.

| Substance | Unit | Simplified | Advanced | Difference [%] |
| --- | --- | --- | --- | --- |
| $CO_2$ | g/PKM | $9.939 \times 10^{1}$ | $9.940 \times 10^{1}$ | 0.01 |
| Crude oil | g/PKM | $3.162 \times 10^{1}$ | $3.162 \times 10^{1}$ | 0.00 |
| Dinitrogen oxide ($NO_2$) | g/PKM | $2.391 \times 10^{-2}$ | $2.398 \times 10^{-2}$ | 0.28 |
| Zinc (Zn) | g/PKM | $1.394 \times 10^{-6}$ | $1.427 \times 10^{-6}$ | 2.31 |
| Arsenic (As) | g/PKM | $2.630 \times 10^{-7}$ | $2.638 \times 10^{-7}$ | 0.29 |
| Dioxine | g/PKM | $2.699 \times 10^{-12}$ | $2.819 \times 10^{-12}$ | 4.45 |
| Lead (Pb) | g/PKM | $5.891 \times 10^{-10}$ | $1.224 \times 10^{-6}$ | >$10^{6}$ |

**Table A4.** *Cont.*

| Substance | Unit | Simplified | Advanced | Difference [%] |
|---|---|---|---|---|
| $O_2$ | g/PKM | $9.627 \times 10^1$ | 0.000 | $-100$ |
| Water (in-flight emission) | g/PKM | $3.483 \times 10^1$ | $3.801 \times 10^1$ | 9.14 |
| $NO_x$ | g/PKM | $4.425 \times 10^{-1}$ | $4.425 \times 10^{-1}$ | 0.00 |
| Hard coal | g/PKM | $1.721 \times 10^{-4}$ | $1.543 \times 10^{-1}$ | $>10^5$ |
| Natural gas | g/PKM | $1.779 \times 10^0$ | $1.780 \times 10^0$ | 0.06 |
| CO | g/PKM | $7.737 \times 10^{-2}$ | 0.000 | $-100$ |
| $SO_2$ | g/PKM | $9.029 \times 10^{-2}$ | $9.027 \times 10^{-2}$ | $-0.02$ |
| HC | g/PKM | $5.077 \times 10^{-3}$ | $5.077 \times 10^{-3}$ | 0.00 |
| $CH_4$ | g/PKM | $9.470 \times 10^{-2}$ | $9.752 \times 10^{-2}$ | 2.98 |
| $N_2O$ | g/PKM | $2.496 \times 10^{-8}$ | $1.845 \times 10^{-4}$ | $>10^6$ |
| PM2.5 | g/PKM | $6.713 \times 10^{-3}$ | $7.104 \times 10^{-3}$ | 5.83 |
| Water (resource) | g/PKM | $6.615 \times 10^{-2}$ | $8.894 \times 10^0$ | $>10^5$ |
| Nickel (Ni) | g/PKM | $1.179 \times 10^{-7}$ | $4.461 \times 10^{-6}$ | $>10^4$ |
| Benzene (C6H6) | g/PKM | 0.000 | $1.067 \times 10^{-5}$ | - |
| Formaldehyde ($CH_2O$) | g/PKM | $2.488 \times 10^{-7}$ | $8.210 \times 10^{-6}$ | $>10^4$ |
| Li | g/PKM | 0.000 | 0.000 | - |
| Phosphate ($PO_4$) | g/PKM | $3.274 \times 10^{-5}$ | $9.752 \times 10^{-5}$ | 197.86 |
| Polycyclic aromatic hydrocarbon (PAH) | g/PKM | $1.327 \times 10^{-7}$ | $2.552 \times 10^{-7}$ | 92.35 |
| Copper ore | g/PKM | 0.000 | $1.879 \times 10^{-5}$ | - |
| H-3 | Bq/PKM | $3.016 \times 10^2$ | $3.206 \times 10^2$ | 6.31 |
| C-14 | Bq/PKM | $1.002 \times 10^{-2}$ | $2.112 \times 10^{-2}$ | 110.81 |
| Cs-134 | Bq/PKM | $1.159 \times 10^{-3}$ | $1.748 \times 10^{-3}$ | 50.79 |
| Cs-137 | Bq/PKM | $7.678 \times 10^{-3}$ | $1.303 \times 10^{-2}$ | 69.72 |
| Co-60 | Bq/PKM | $2.915 \times 10^{-3}$ | $5.424 \times 10^{-3}$ | 86.06 |
| Rn-222 | Bq/PKM | $6.428 \times 10^0$ | $1.270 \times 10^1$ | 97.51 |

The following list summarizes the specific explanation with respect to differences:

- *Lead*, *hard coal*, *$N_2O$* and *PAH*: The *simplified LCA, lead* is only mentioned in the processes "steel production" and consequently "test flight campaign" (acc. to its inventory analysis, 4% SS contribution with $5.89 \times 10^{-10}$ g/PKM). The emission also occurs in the "kerosene production" with a 0.08% process-SS share and was therefore neglected. In the context of the overall process in the *advanced LCA*, the contribution is higher than 1% and should not be neglected. Similar can be applied to *hard coal*, *$N_2O$* and *PAH*.
- *Water (in-flight emission)*: Emissions are considered contributing to the Contrails and Cirrus Clouds formation and the global warming potential acc. to the climate model.
- *Water (resource)*: water is only considered in CFRP production in the *simplified LCA* whereas, in *advanced*, the highest water consumption comes from kerosene production, which is now included in the LCIA. Additionally, only categories "water, surface" and "water, river" are considered relevant to keep comparability.
- *Ni* and *$PO_4$*: The *simplified LCA* only considers *Nickel* in the process "use of production facilities" whereas in the *advanced LCA*, "kerosene production" also contributes to a higher absolute number. Similar applies to *$PO_4$* Additionally, for *$PO_4$*, only emissions to water are considered, and to soil are neglected due to a lower CF.
- *Formaldehyde*: the *simplified LCA* only considers it in the processes "computer usage" and "material production (composites)", whereas in the *advanced LCA*, "kerosene production" contributes to a higher absolute number.
- *C-14*, *C-137*, *Cs-134*, *Co-60*, *Rn-222*: The *simplified LCA* only considers emissions in the processes "computer usage", "use of production facilities", "material production (composites)" and "test flights" whereas in the *advanced LCA*, "kerosene production" is also added. Additionally, for *C-14* in both methodologies, only "emissions to air" is included in the table (CF is higher compared to "emissions to water, unspecified and fresh water"). Also, for *C-137*, only emissions to water is shown.

*Appendix B.3. Summarized Life Cycle Assessment Results*

**Table A5.** Results of life cycle impact assessment (simplified (bugfixed) vs. advanced LCA) of the kerosene-powered concept.

| Mid-/Endpoint Categories | Unit | Value (Simplified) | Value (Advanced) | Difference [%] |
|---|---|---|---|---|
| Climate change | g $CO_2$ eq | $3.287 \times 10^2$ | $3.289 \times 10^2$ | 0.05 |
| Ozone depletion | g CFC-11 eq | $2.746 \times 10^{-10}$ | $2.169 \times 10^{-6}$ | $> 10^6$ |
| Particulate matter formation | g PM2.5 eq | $8.420 \times 10^{-2}$ | $8.461 \times 10^{-2}$ | 0.49 |
| Ionizing radiation | Bq Co-60 eq | $1.277 \times 10^{-1}$ | $2.221 \times 10^{-1}$ | 73.94 |
| Human carcinogenic toxicity | g 1.4-DCB eq | $1.409 \times 10^{-3}$ | $5.173 \times 10^{-3}$ | $> 10^2$ |
| Human non-carcinogenic toxicity | g 1.4-DCB eq | $4.326 \times 10^{-2}$ | $6.523 \times 10^{-1}$ | $> 10^3$ |
| Photochemical oxidant formation (HH) | g $NO_x$ eq | $4.672 \times 10^{-1}$ | $4.692 \times 10^{-1}$ | 0.42 |
| Photochemical oxidant formation (ECO) | g $NO_x$ eq | $4.677 \times 10^{-1}$ | $4.708 \times 10^{-1}$ | 0.67 |
| Water use | g consumed | $6.615 \times 10^{-2}$ | 8.944 | $> 10^4$ |
| Terrestrial acidification | g $SO_2$ eq | $2.582 \times 10^{-1}$ | $2.584 \times 10^{-1}$ | 0.07 |
| Freshwater eutrophication | g P eq | $1.080 \times 10^{-5}$ | $6.205 \times 10^{-5}$ | $> 10^2$ |
| Marine eutrophication | g N eq | 0.000 | $1.205 \times 10^{-4}$ | 0.00 |
| Terrestrial ecotoxicity | g 1.4-DCB eq | $2.207 \times 10^{-1}$ | 7.333 | $> 10^3$ |
| Freshwater ecotoxicity | g 1.4-DCB eq | $2.204 \times 10^{-2}$ | $3.605 \times 10^{-3}$ | $-83.64$ |
| Marine ecotoxicity | g 1.4-DCB eq | $1.840 \times 10^{-4}$ | $8.513 \times 10^{-2}$ | $> 10^5$ |
| Land use | m2/a | 0.000 | $3.106 \times 10^{-4}$ | 0.00 |
| Mineral resource scarcity | g Fe eq | 0.000 | $2.044 \times 10^{-3}$ | 0.00 |
| Fossil resource scarcity | g oil eq | $3.370 \times 10^1$ | $3.377 \times 10^1$ | 0.22 |
| Damage to human health | DALY | $3.574 \times 10^{-7}$ | $3.580 \times 10^{-7}$ | 0.16 |
| Damage to ecosystems | species.yr | $1.033 \times 10^{-9}$ | $1.036 \times 10^{-9}$ | 0.37 |
| Damage to resource availability | USD2013 | $1.508 \times 10^{-2}$ | $1.519 \times 10^{-2}$ | 0.79 |
| Single Score | points | $8.387 \times 10^{-2}$ | $8.554 \times 10^{-2}$ | 1.99 |

**Table A6.** Results of life cycle impact assessment (simplified (bugfixed) vs. advanced LCA) of the hydrogen-powered concept (steam reforming).

| Mid-/Endpoint Categories | Unit | Value (Simplified) | Value (Advanced) | Difference [%] |
|---|---|---|---|---|
| Climate change | g $CO_2$ eq | $8.427 \times 10^2$ | $8.437 \times 10^2$ | 0.12 |
| Ozone depletion | g CFC-11 eq | $2.522 \times 10^{-10}$ | $5.157 \times 10^{-5}$ | $>10^7$ |
| Particulate matter formation | g PM2.5 eq | $3.197 \times 10^{-1}$ | $3.398 \times 10^{-1}$ | 6.30 |
| Ionizing radiation | Bq Co-60 eq | $1.510 \times 10^1$ | $1.839 \times 10^1$ | 21.81 |
| Human carcinogenic toxicity | g 1.4-DCB eq | $3.546 \times 10^{-2}$ | $8.086 \times 10^{-2}$ | $>10^2$ |
| Human non-carcinogenic toxicity | g 1.4-DCB eq | $3.356 \times 10^{-2}$ | 2.700 | $>10^3$ |
| Photochemical oxidant formation (HH) | g $NO_x$ eq | $4.900 \times 10^{-1}$ | $4.912 \times 10^{-1}$ | 0.25 |
| Photochemical oxidant formation (ECO) | g $NO_x$ eq | $4.905 \times 10^{-1}$ | $4.926 \times 10^{-1}$ | 0.43 |
| Water use | g consumed | $6.609 \times 10^{-2}$ | $1.003 \times 10^2$ | $>10^5$ |
| Terrestrial acidification | g $SO_2$ eq | $1.070 \times 10^0$ | $1.070 \times 10^0$ | 0.05 |
| Freshwater eutrophication | g P eq | $1.080 \times 10^{-5}$ | $3.309 \times 10^{-5}$ | $>10^2$ |
| Marine eutrophication | g N eq | 0.000 | $3.404 \times 10^{-4}$ | 0.00 |
| Terrestrial ecotoxicity | g 1.4-DCB eq | $4.396 \times 10^{-2}$ | $3.867 \times 10^1$ | $>10^4$ |
| Freshwater ecotoxicity | g 1.4-DCB eq | $2.203 \times 10^{-2}$ | $7.332 \times 10^{-3}$ | $-66.72$ |
| Marine ecotoxicity | g 1.4-DCB eq | $8.773 \times 10^{-5}$ | $3.985 \times 10^{-2}$ | $>10^4$ |
| Land use | m2/a | 0.000 | $3.106 \times 10^{-4}$ | 0.00 |
| Mineral resource scarcity | g Fe eq | 0.000 | $1.389 \times 10^{-3}$ | 0.00 |
| Fossil resource scarcity | g oil eq | $1.070 \times 10^1$ | $3.220 \times 10^1$ | $>10^2$ |
| Damage to human health | DALY | $9.752 \times 10^{-7}$ | $9.895 \times 10^{-7}$ | 1.46 |
| Damage to ecosystems | species.yr | $2.624 \times 10^{-9}$ | $2.632 \times 10^{-9}$ | 0.30 |
| Damage to resource availability | USD2013 | $3.252 \times 10^{-3}$ | $7.005 \times 10^{-3}$ | 115.42 |
| Single Score | points | $1.784 \times 10^{-1}$ | $1.887 \times 10^{-1}$ | 5.77 |

**Table A7.** Results of life cycle impact assessment (simplified (bugfixed) vs. advanced LCA) of hydrogen-powered concept (electrolysis).

| Mid-/Endpoint Categories | Unit | Value (Simplified) | Value (Advanced) | Difference [%] |
|---|---|---|---|---|
| Climate change | g $CO_2$ eq | $4.711 \times 10^2$ | $4.713 \times 10^2$ | 0.05 |
| Ozone depletion | g CFC-11 eq | $2.522 \times 10^{-10}$ | $5.512 \times 10^{-7}$ | $> 10^5$ |
| Particulate matter formation | g PM2.5 eq | $2.000 \times 10^{-2}$ | $2.072 \times 10^{-2}$ | 3.59 |
| Ionizing radiation | Bq Co-60 eq | $1.278 \times 10^{-1}$ | $1.987 \times 10^{-1}$ | 55.43 |
| Human carcinogenic toxicity | g 1.4-DCB eq | $4.876 \times 10^{-3}$ | $5.962 \times 10^{-3}$ | 22.27 |
| Human non-carcinogenic toxicity | g 1.4-DCB eq | $1.420 \times 10^{-1}$ | $2.050 \times 10^{-1}$ | 44.37 |
| Photochemical oxidant formation (HH) | g $NO_x$ eq | $5.437 \times 10^{-2}$ | $5.410 \times 10^{-2}$ | $-0.49$ |
| Photochemical oxidant formation (ECO) | g $NO_x$ eq | $5.437 \times 10^{-2}$ | $5.418 \times 10^{-2}$ | $-0.35$ |
| Water use | g consumed | $1.582 \times 10^1$ | $1.601 \times 10^{-5}$ | 1.19 |
| Terrestrial acidification | g $SO_2$ eq | $4.478 \times 10^{-2}$ | $4.692 \times 10^{-2}$ | 4.78 |
| Freshwater eutrophication | g P eq | $1.080 \times 10^{-5}$ | $1.211 \times 10^{-5}$ | 12.11 |
| Marine eutrophication | g N eq | 0.000 | $1.977 \times 10^{-5}$ | 0.00 |
| Terrestrial ecotoxicity | g 1.4-DCB eq | $3.097 \times 10^{-1}$ | $1.436 \times 10^0$ | $> 10^2$ |
| Freshwater ecotoxicity | g 1.4-DCB eq | $2.203 \times 10^{-2}$ | $8.778 \times 10^{-3}$ | $-60.16$ |
| Marine ecotoxicity | g 1.4-DCB eq | $5.941 \times 10^{-5}$ | $2.812 \times 10^{-3}$ | $> 10^3$ |
| Land use | m2/a | 0.000 | $3.106 \times 10^{-4}$ | 0.00 |
| Mineral resource scarcity | g Fe eq | 0.000 | $4.230 \times 10^{-3}$ | 0.00 |
| Fossil resource scarcity | g oil eq | $5.420 \times 10^{-1}$ | $9.358 \times 10^{-1}$ | 72.65 |
| Damage to human health | DALY | $4.413 \times 10^{-7}$ | $4.420 \times 10^{-7}$ | 0.15 |
| Damage to ecosystems | species.yr | $1.310 \times 10^{-9}$ | $1.314 \times 10^{-9}$ | 0.29 |
| Damage to resource availability | USD2013 | $2.222 \times 10^{-4}$ | $2.852 \times 10^{-4}$ | 28.37 |
| Single Score | points | $7.309 \times 10^{-2}$ | $7.349 \times 10^{-2}$ | 0.55 |

**Table A8.** Results of life cycle impact assessment (simplified (bugfixed) vs. advanced LCA) of the battery-electric aircraft (EU mix).

| Mid-/Endpoint Categories | Unit | Value (Simplified) | Value (Advanced) | Difference [%] |
|---|---|---|---|---|
| Climate change | g $CO_2$ eq | $1.212 \times 10^2$ | $1.221 \times 10^2$ | 0.74 |
| Ozone depletion | g CFC-11 eq | $7.816 \times 10^{-10}$ | $4.705 \times 10^{-5}$ | $>10^6$ |
| Particulate matter formation | g PM2.5 eq | $2.377 \times 10^{-1}$ | $2.539 \times 10^{-1}$ | 6.85 |
| Ionizing radiation | Bq Co-60 eq | $1.585 \times 10^1$ | $1.683 \times 10^1$ | 6.18 |
| Human carcinogenic toxicity | g 1.4-DCB eq | $3.030 \times 10^{-2}$ | $4.237 \times 10^{-2}$ | 39.86 |
| Human non-carcinogenic toxicity | g 1.4-DCB eq | $9.274 \times 10^{-1}$ | $2.454 \times 10^0$ | 164.60 |
| Photochemical oxidant formation (HH) | g $NO_x$ eq | $2.201 \times 10^{-1}$ | $2.123 \times 10^{-1}$ | $-3.53$ |
| Photochemical oxidant formation (ECO) | g $NO_x$ eq | $2.201 \times 10^{-1}$ | $2.131 \times 10^{-1}$ | $-3.18$ |
| Water use | g consumed | $1.883 \times 10^{-1}$ | $9.132 \times 10^{-5}$ | $>10^4$ |
| Terrestrial acidification | g $SO_2$ eq | $7.690 \times 10^{-1}$ | $7.626 \times 10^{-1}$ | $-0.84$ |
| Freshwater eutrophication | g P eq | $2.161 \times 10^{-5}$ | $4.225 \times 10^{-5}$ | 95.50 |
| Marine eutrophication | g N eq | 0.000 | $3.275 \times 10^{-4}$ | 0.00 |
| Terrestrial ecotoxicity | g 1.4-DCB eq | $3.915 \times 10^{-1}$ | $3.535 \times 10^1$ | $>10^3$ |
| Freshwater ecotoxicity | g 1.4-DCB eq | $4.408 \times 10^{-2}$ | $7.166 \times 10^{-3}$ | $-83.75$ |
| Marine ecotoxicity | g 1.4-DCB eq | $3.025 \times 10^{-4}$ | $3.741 \times 10^{-2}$ | $>10^4$ |
| Land use | m2/a | 0.000 | $6.212 \times 10^{-4}$ | 0.00 |
| Mineral resource scarcity | g Fe eq | $2.859 \times 10^0$ | $2.761 \times 10^0$ | $-3.44$ |
| Fossil resource scarcity | g oil eq | $1.367 \times 10^1$ | $2.994 \times 10^1$ | $>10^2$ |
| Damage to human health | DALY | $2.626 \times 10^{-7}$ | $2.740 \times 10^{-7}$ | 4.34 |
| Damage to ecosystems | species.yr | $5.307 \times 10^{-10}$ | $5.381 \times 10^{-10}$ | 1.40 |
| Damage to resource availability | USD2013 | $5.346 \times 10^{-3}$ | $7.269 \times 10^{-3}$ | 35.99 |
| Single Score | points | $5.721 \times 10^{-2}$ | $6.188 \times 10^{-2}$ | 8.16 |

**Table A9.** Results of life cycle impact assessment (simplified (bugfixed) vs. advanced LCA) of the battery-electric aircraft (renewable energies).

| Mid-/Endpoint Categories | Unit | Value (Simplified) | Value (Advanced) | Difference [%] |
|---|---|---|---|---|
| Climate change | g $CO_2$ eq | $1.058 \times 10^1$ | $1.082 \times 10^1$ | 2.32 |
| Ozone depletion | g CFC-11 eq | $7.816 \times 10^{-10}$ | $5.372 \times 10^{-7}$ | $>10^4$ |
| Particulate matter formation | g PM2.5 eq | $2.910 \times 10^{-2}$ | $2.933 \times 10^{-2}$ | 0.80 |
| Ionizing radiation | Bq Co-60 eq | $2.645 \times 10^{-1}$ | $2.829 \times 10^{-1}$ | 6.93 |
| Human carcinogenic toxicity | g 1.4-DCB eq | $1.688 \times 10^{-3}$ | $2.210 \times 10^{-3}$ | 30.93 |
| Human non-carcinogenic toxicity | g 1.4-DCB eq | $3.284 \times 10^{-2}$ | $8.926 \times 10^{-2}$ | $>10^2$ |
| Photochemical oxidant formation (HH) | g $NO_x$ eq | $8.612 \times 10^{-3}$ | $8.694 \times 10^{-3}$ | 0.95 |
| Photochemical oxidant formation (ECO) | g $NO_x$ eq | $8.612 \times 10^{-3}$ | $8.744 \times 10^{-3}$ | 1.53 |
| Water use | g consumed | $2.996 \times 10^0$ | $3.377 \times 10^{-6}$ | 12.70 |
| Terrestrial acidification | g $SO_2$ eq | $5.387 \times 10^{-2}$ | $5.434 \times 10^{-2}$ | 0.88 |
| Freshwater eutrophication | g P eq | $2.161 \times 10^{-5}$ | $2.253 \times 10^{-5}$ | 4.26 |
| Marine eutrophication | g N eq | 0.000 | $3.444 \times 10^{-5}$ | 0.00 |
| Terrestrial ecotoxicity | g 1.4-DCB eq | $1.317 \times 10^{-1}$ | $6.973 \times 10^{-1}$ | $>10^2$ |
| Freshwater ecotoxicity | g 1.4-DCB eq | $4.407 \times 10^{-2}$ | $2.419 \times 10^{-3}$ | −94.51 |
| Marine ecotoxicity | g 1.4-DCB eq | $6.701 \times 10^{-5}$ | $2.680 \times 10^{-3}$ | $> 10^3$ |
| Land use | m2/a | 0.000 | $6.212 \times 10^{-4}$ | 0.00 |
| Mineral resource scarcity | g Fe eq | $2.859 \times 10^0$ | $2.760 \times 10^0$ | 0.00 |
| Fossil resource scarcity | g oil eq | $1.035 \times 10^0$ | $1.193 \times 10^0$ | 15.20 |
| Damage to human health | DALY | $2.815 \times 10^{-8}$ | $2.852 \times 10^{-8}$ | 1.31 |
| Damage to ecosystems | species.yr | $4.221 \times 10^{-11}$ | $4.854 \times 10^{-11}$ | 15.01 |
| Damage to resource availability | USD2013 | $1.033 \times 10^{-3}$ | $1.078 \times 10^{-3}$ | 4.36 |
| Single Score | points | $3.551 \times 10^{-3}$ | $3.768 \times 10^{-3}$ | 6.12 |

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
