# Peer review of "Integrating Life Cycle Assessment in Conceptual Aircraft Design: A Comparative Tool Analysis"

_aerospace, doi:10.3390/aerospace11010101_

Round 1

Reviewer 1 Report

Comments and Suggestions for Authors

Dear authors,

thanks for submitting your work. Your article is overall valuable and focuses on a crucial aspect, often disregarded by many working in the aircraft conceptual design phase. The integration of tools for a preliminary life-cycle analysis is fundamental, especially when considering breakthrough technologies, such as new materials for the airframe or advanced propulsion concepts. Your work can help many aircraft designers focusing on the conceptual/preliminary design phase in making this further step towards a more comprehensive vision of the aircraft system and its impact on the environment.

Please, consider my following comments as a suggestion to further improve your manuscript:

1) Regarding section 2.1, at least according to my experience, inventories can be also setup in collaboration with an industrial partner, if present/available. In this case, required inventory data can be filled by including information provided directly by people working in contact with the production chain.

2) Regarding section 2.2, are there more examples of comparative analyses between different LCA tools? Please, if available, add a few more.

3) At line 193, could you clarify what Umberto is? Because there is no reference given.

4) Regarding section 2.4, I would have preferred a more personal review of previous works, with a little more details on the case studies. But citing the most recent literature reviews might be sufficient as well.

5) As regards section 3.1, I assume that more details about the 5 aircraft models considered by [12, 13] are given by those references. However, I would consider providing some basic information regarding their characteristics and design criteria in your article, to ease the reader. 

6) Regarding section 3.2, although I overall agree with the priorities you gave, I would argue that they are something really subjective and linked to the personal experience of the user. They may also depend on the type of study/analysis to be performed.

7) Still regarding section 3.2, since in the last paragraph you provide your answer to the first of the research question, I would have expected a few more explanations on why you did select a specific degree of priority for a certain qualitative/quantitative metric. I would try to elaborate a little bit more on this.

8) As regards section 5.1, for energy carrier production do you just mean the production of kerosene/hydrogen/electricity for recharging batteries, right?

9) Still regarding section 5.1, for me it's not clear where the discrepancies that you highlight between the simplified and the advanced tools originate from. I assume the explanation is, at least in part, the one given in the Appendix. Consider providing a summary on those considerations also in this section. Nevertheless, I still have some questions regarding the hypotheses you made using the advanced tool: are you considering the same LCIA method and perspective used by the simplified one, for a fair comparison? Consider adding these explanations.

10) Please explain what DALY stands for when first introducing it (I see it is explained in the abbreviations).

11) Regarding table A1, for me it's not clear which differences are reported. Please clarify.

Comments on the Quality of English Language

Concerning the quality of English, please consider the following comments to improve your manuscript:

1) Check the sentence at lines 94-96, as it does not make too much sense to me.

2) Similarly, check the punctuation and the sentence in general at lines 192-193.

3) Check the sentence at lines 268-269.

4) Check the sentence at line 374-376.

Author Response

Dear reviewer,

thank you for your time and valuable feedback. Attached are my detailed comments in your review on how I implemented them.

Best regards,

Kristina Mazur

Reviewer 2 Report

Comments and Suggestions for Authors

Very good paper, written clearly and concisely. Kudos to the authors.

Author Response

Dear reviewer, thank you very much for the kind words and your review!